# Effect of Citrate- and Gold-Stabilized Superparamagnetic Iron Oxide Nanoparticles on Head and Neck Tumor Cell Lines during Combination Therapy with Ionizing Radiation

**DOI:** 10.3390/bioengineering9120806

**Published:** 2022-12-15

**Authors:** Christoph Schreiber, Tim Franzen, Laura Hildebrand, René Stein, Bernhard Friedrich, Rainer Tietze, Rainer Fietkau, Luitpold V. Distel

**Affiliations:** 1Department of Radiation Oncology, University Hospital Erlangen, Friedrich-Alexander-University Erlangen-Nürnberg, 91054 Erlangen, Germany; 2Comprehensive Cancer Center Erlangen-EMN (CCC ER-EMN), 91054 Erlangen, Germany; 3ENT-Department, Else Kröner-Fresenius-Stiftung Professorship, Section for Experimental Oncology and Nanomedicine (SEON), University Hospital Erlangen, 91054 Erlangen, Germany

**Keywords:** head and neck cancer cell lines, nanoparticles, superparamagnetic iron oxide nanoparticles, ionizing radiation, interactions, citrate, gold

## Abstract

Head and neck squamous cell carcinoma (HNSCC) is the sixth most common cancer worldwide. They are associated with alcohol and tobacco consumption, as well as infection with human papillomaviruses (HPV). Therapeutic options include radiochemotherapy, surgery or chemotherapy. Nanoparticles are becoming more and more important in medicine. They can be used diagnostically, but also therapeutically. In order to provide therapeutic alternatives in the treatment of HNSCC, the effect of citrate-coated superparamagnetic iron oxide nanoparticles (Citrate-SPIONs) and gold-coated superparamagnetic iron oxide nanoparticles (Au-SPIONs) in combination with ionizing irradiation (IR) on two HPV positive and two HPV negative HNSCC and healthy fibroblasts and keratinocytes cell lines were tested. Effects on apoptosis and necrosis were analyzed by using flow cytometry. Cell survival studies were performed with a colony formation assay. To better understand where the SPIONs interact, light microscopy images and immunofluorescence studies were performed. The HNSCC and healthy cell lines showed different responses to the investigated SPIONs. The cytotoxic effects of SPIONs, in combination with IR, are dependent on the type of SPIONs, the dose administered and the cell type treated. They are independent of HPV status. Reasons for the different cytotoxic effect are probably the different compositions of the SPIONs and the related different interaction of the SPIONs intracellularly and paramembranously, which lead to different strong formations of double strand breaks.

## 1. Introduction

Head and neck squamous cell carcinomas (HNSCC) are the sixth most common cancer worldwide [1]. They are localized in the oral cavity, pharynx and larynx. In general, they are often associated with tobacco consumption, alcohol abuse and an infection with human papillomavirus (HPV) [1,2,3]. Because of the continually increasing consumption of alcohol worldwide and the rising trend in HPV-related head and neck cancers, the treatment of HNSCC is of great importance [4,5]. Common treatment is combined radiochemotherapy (RCT), surgery and chemotherapy [6,7,8,9]. Chemotherapies in particular often have a number of side effects. These include diarrhea or nausea, for example, but serious complications such as kidney toxicity or bone marrow toxicity also occur [10,11]. Surgical interventions also present risks. These include, for example. the possibility of perioperative blood loss, infection or the occurrence of wound healing problems after surgery [12,13,14]. All in all, patients diagnosed with HNSCC have a worse quality of life [15]. Therefore, it is even more important that there are multiple diagnostic and treatment options for this group of carcinomas. One of the latest methods is the use of synthesized nanoparticles.

In tumor diagnostics and treatment, nanoparticles have recently become more important and function as diagnostic markers [16,17,18] and drug delivery devices [19,20,21,22,23]. There are several trials in which it has been shown that nanoparticles help detecting tumor tissue via photoimaging [17,24] and can deliver chemotherapeutics to the tumor sites [25,26]. However, it is also of interest whether nanoparticles alone harm tumor cells and normal cell tissue. The research focuses on the effect of gold-coated superparamagnetic iron oxide nanoparticles (Au-SPIONs) and citrate-coated superparamagnetic iron oxide nanoparticles (Citrate-SPIONs) to HNSCC tumor cells.

In general, SPIONs consist of ferrimagnetic iron oxides such as magnetite or maghemite. Condensed into nanoparticles with sizes of less than 20 nm, these materials exhibit superparamagnetic behavior [27,28]. This is important for the application of SPIONs in patients to prevent magnetic agglomeration of SPIONs in arteries.

SPIONs are stabilized on their surfaces to hinder nanoparticle agglomeration due to their high surface area. One of the simplest surface stabilizations is achieved by using citrate ions, which can bind covalently to the iron oxide surface through their carboxyl groups (Figure 1A) [29,30]. Enhanced functionality of the SPION surface can be achieved by a gold coating on the SPIONs (Au-SPIONs, Figure 1B), which can be easily functionalized with thiol-containing molecules [31,32]. This facile modification motif makes Au-SPIONs promising candidates e.g., for clinical use. To ensure a possible clinical use of SPIONs with ionizing irradiation (IR) it is important to know the biological behavior of the SPIONs. In particular, with regard to the cytotoxicity of the nanoparticles themselves and whether they cause damage in combination with IR. In this work, the effect of Au-SPIONs and Citrate-SPIONs on four different HNSCC tumor cell lines were studied. In addition, two healthy cell lines were also studied to determine if there were also effects on normal tissue cells. Cell death by apoptosis and necrosis and cell inactivation was primarily studied by the colony formation assay to score the effects on the survival fraction of tumor cells and normal tissue cells.

## 2. Materials and Methods

### 2.1. Cell Culture

Four different HNSCC, primary human fibroblasts and healthy keratinocytes were used to study the influence of different SPIONs systems in combination with ionizing radiation. The HNSCC Cal33 and HSC4 are HPV-16 negative (HPV-), whereas UM-SCC-47 and UD-SCC-2 are HPV-16 positive (HPV+) [33]. Samples were obtained courtesy of Dr. Thorsten Rieckmann from the University of Medical Centre Hamburg-Eppendorf, Germany. Primary human skin fibroblasts SBLF9 were derived from a healthy young Caucasian men. Healthy keratinocytes HaCAT were derived from healthy individuals and subsequently cultured. They were obtained from the Department of Ophtalmology, University Clinic Erlangen [34].

Fibroblasts were grown in F-12 medium (Gibco, Waltham, MA, USA) with 15% fetal bovine serum (FBS, Biochrom, Berlin, Germany), 5% penicillin-streptomycin (Gibco, Waltham, MA, USA), and 5% non-essential amino acids (NEA, Biochrom, Berlin, Germany). Keratinocytes were cultured in Dulbecco’s MEM (DMEM, Gibco, Waltham, MA, USA) with 4.5 g/L, 10% FBS and 1% penicillin-streptomycin. HNSCC were cultured in Dulbecco’s MEM (DMEM, Gibco, Waltham, MA, USA) supplemented with 10% FBS and 1% penicillin-streptomycin. All cells were incubated at a constant temperature of 37 °C and in a humidified atmosphere with 5% CO_2_.

### 2.2. Citrate-SPIONs and Au-SPIONs

Citrate-stabilized superparamagnetic iron oxide nanoparticles were synthesized following the protocol of Stein et al. [35]. Briefly, SPION precipitation was achieved by a rapid injection of a 25% ammonia solution into a vigorously stirring aqueous solution of iron (II) and iron (III) salts containing 13.2 mg iron/mL under inert gas protection. After growing the SPIONs for 10 min at room temperature, 15 mL of a 293.3 mg/mL sodium citrate solution was injected and the temperature was raised to 90 °C for 30 min to complete the surface stabilization of the SPIONs due to citrate ions. Excess citrate was removed by dialyzing the cooled Citrate-SPIONs against H_2_O for 6 h.

Citrate-SPIONs were coated with gold following the procedure of Elbialy et al. [36] with adaptations made by Stein et al. [35] In short, Citrate-SPIONs were diluted in a volume of 35 mL to a concentration of 1 mg Fe/mL using H_2_O. The dispersion was brought to a boil in an inert atmosphere. After addition of a 35.3 mM HAuCl_4_ solution, the dispersion turned deeply red due to the coating of Citrate-SPIONs with gold (Au-SPIONs). To further stabilize the Au-SPIONs, a 1 mg/mL sodium citrate solution was added after 15 min of stirring while still boiling the dispersion.

Characterization of both SPION systems was performed similarly to the protocols used by Mühlberger et al. [37]. The hydrodynamic size of the particle systems was measured to be 107 ± 3 nm with a corresponding PDI value of 0.197 ± 0.003 and 148 ± 10 nm with a corresponding PDI value of 0.184 ± 0.004 for Citrate-SPIONs and Au-SPIONs, respectively, using dynamic light scattering (Zetasizer Nano, Malvern instruments, Worcestershire, UK) as analysis. At pH 7 the Citrate-SPIONs exhibited a ζ-potential of −30 ± 1 mV, while Au-SPIONs showed a value of −49 ± 1 mV. For identification of the particle amounts used during biological experiments, the iron and gold content of the SPION systems was analyzed using microwave induced plasma atomic emission spectroscopy (Agilent 4200 MP-AES, Agilent Technologies, Santa Clara, CA, USA).

### 2.3. Flow Cytometry Analysis of Apoptosis and Necrosis

HNSCC and healthy tissue cells (SBLF9 and HaCAT) were seeded in T25 flasks. The numbers of cells were chosen between 20,000 and 50,000 cells, depending on the proliferation rates of the cell lines. The cells were cultured in 1.5 mL cell culture medium for 48 h. Afterwards, the flasks were divided into six groups. The first group was untreated and used as control. The second group was treated with 10 µg/mL Citrate-SPIONs and the third group with 2.5 µg/mL Au-SPIONs in 1.5 mL medium. The doses given are theoretical doses, since the nanoparticles are not homogeneously distributed, but sink to the cells, causing a higher concentration locally on the cells. The layer thickness above the cells was always kept constant at 1.5 mm, so that the same concentration always occurred at the cells when the theoretical concentration was added. Group four was irradiated with 2 Gy ionizing radiation by an ISOVOLT Titan X-ray generator (GE, Ahrensburg, Germany). The last two groups were firstly treated with 10 µg/mL Citrate-SPIONs and 2.5 µg/mL Au-SPIONs in 1.5 mL medium and after 3 h irradiated with 2 Gy. Subsequently the flasks were incubated at 37 °C and 5% CO_2_ for 48 h. Afterwards, the cells were harvested and centrifuged (8 min, 20 °C, 300× *g*). The supernatants were removed and 200 µL cold Ringer’s solution (Fresenius Kabi, Bad Homburg, Germany) plus 10 µL of a mixture of APC Annexin and 7AAD (BD Biosciences, Franklin Lakes, NJ, USA) was added. The cells were kept on ice and incubated with light protection for 30 min. Subsequently the cells were centrifuged (6 min, 20 °C, 400× *g*), the staining agent was removed and the cells were prepared for analyzing with a CytoFLEX S flow cytometer (Beckmann Coulter, Brea, CA, USA). Kaluza analysis software (Beckman Coulter, Brea, CA, USA) was used to analyze apoptosis and necrosis rates.

### 2.4. Gating Strategy for Flow Cytometry

Forward and side scattering of the flow cytometer was used to distinguish cells from other particles. The cellular granularity was measured by the side scatter, while the size of cells was identified by the forward scatter. The area with cells was gated and used for apoptosis/necrosis analyses. Cells stained by Annexin APC classified as apoptotic cells and positive cells for both (Annexin APC and 7AAD) were assigned as necrotic cells. Cells with no staining (Annexin APC-neg. and 7AAD-neg.) were defined as viable cells.

### 2.5. Colony Formation Assay

Malignant and healthy cells were seeded in 6-well petri dishes. After 24 h of incubation, every cell line was divided into 10 groups. The first group were untreated and used as control group. The second one was added with 2.5 µg/mL Au-SPIONs or 40 µg/mL Citrate-SPIONs in 1.5 mL medium and the other groups were irradiated with 1, 2, 4 and 6 Gy IR or irradiated with IR plus SPIONs. The number of seeded cells depended on the dose of IR and proliferation rates of the cell lines.

After 72 h of treatment the medium was exchanged with drug-free medium. Now the cells were incubated at 37 °C and 5% CO_2_ until they reached >50 cells of each colony. Afterwards the dishes were stained with methylene blue (#66725, Sigma Aldrich, München, Germany) by room temperature for 30 min. After this, dishes were removed from staining solution and images of stained dishes were acquired by a self-built optic and a high-resolution camera (uEye UI-3180CP, IDS, Obersulm, Germany). Colonies consisting of at least 50 cells were counted by Biomas analyzing software. The rates of colony size (CS) and survival fraction (SF) were plotted in semi logarithmic graphs.

### 2.6. Immunofluorescence Microscopy

Cells were cultured on slides (up to 150,000 cells) for analyzing cell proliferation and DNA damage. Forty-eight hours later, the medium was exchanged, cells were treated with 2.5 µg/mL Au-SPIONs or 40 µg/mL Citrate-SPIONs in 1.5 mL medium and irradiated with 2 Gy after three hours. After treatment, the cells were incubated for 24 h at 37 °C. Then cells were fixed and permeabilized with 4% formaldehyde and 0.1% Triton X-100/PBS for 15 min and afterwards blocked with 1% BSA for 1 h at room temperature. For staining, the primary antibody mouse anti-*γ*H2Ax (1:2500, Merck, Darmstadt, Germany) and the secondary antibody anti-rabbit Ki-67 (1:700, Abcam, Cambridge, UK) were used. DAPI was then added to stain the nucleus. Thereafter, Vectashield (Vector Laboratories, Burlingame, CA, USA) was applied to the slide and covered with a cover slip. Images were taken with a Zeiss Imager Z2 fluorescence microscope (Zeiss, Oberkochen, Germany). Electromagnetic convergence points (foci) generated with anti-*γ*H2Ax were quantified using Biomas software.

### 2.7. Images of Treated Cells

Cal33 and SBLF9 were seeded in T25 flasks. The cells were incubated for 24 h at 37 °C and 5% CO_2_. Subsequently, the malignant and healthy cell lines were added with 2.5 µg/mL Au-SPIONs or 40 µg/mL Citrate-SPIONs in 1.5 mL medium from 2.1 and a control group without SPION’s. Images were acquired after 3, 24, 48 and 72 h of treatment with Zeiss Primo Vert and Leica DM IL Fluo microscope.

### 2.8. Statistics

For statistics and graphs, GraphPad Prism 8 Software was used (GraphPad Software, San Diego, CA, USA). Data were analyzed by unpaired, one-tailed Mann–Whitney U-test. The experiments were repeated at least three times.

## 3. Results

The effect of Au-SPIONs and Citrate-SPIONs (Figure 1) in combination with ionizing radiation on HNSCC lines as well as on normal skin fibroblasts SBLF9 and the keratinocyte cell line HACAT was studied to determine whether the combination of treatments had a combinatorial effect.

### 3.1. Adhesion of SPIONs to the Cells

Images of the cells under a light microscope with the addition of Citrate-SPIONs and Au-SPIONs were taken to better understand the effect of SPIONs. Of interest was whether the SPIONs, when only in the medium, adhered to the cell membrane or were taken up into the cytoplasm. As a proxy, the HNSCC cell line Cal33 (Figure 2) and the fibroblasts SBLF9 (Figure 3) were used because of their large cell bodies. Although nanoparticles with a size of around 10 nm cannot be visualized under the light microscope, the property that the nanoparticles precipitate in the medium and aggregate into larger clusters was exploited so that they could be imaged under the microscope.

Images of treated cells were acquired 3 h, 24 h, 48 h and 72 h after the addition of SPIONs at 630× magnification. In contrast to the cells treated with Au-SPIONs, the cultures with Citrate-SPIONs were added at a higher dose so that more nanoparticles were visible in the medium. Cal33 cells treated with Au-SPIONs show a ring (fringe) of aggregated nanoparticles around the colony from 72 h onwards. Individual clumps are also detectable intracellularly. Cells treated with Citrate-SPIONs no longer show a clearly demarcated cell membrane after 3.5 h, indicating attachment of the nanoparticles. After 48 h, intracellular SPIONs can be assumed to be present (Figure 2). SPIONs were also incorporated into SBLF9 fibroblasts after 24 h. (Figure 3). In particular, in cells similar to Cal33 cells, higher doses of Citrate-SPIONs result in cell membranes that are no longer clearly demarcated. This suggests that nanoparticles tend to adhere to the cell membrane, especially at higher doses.

### 3.2. Determination of the SPIONs Concentration

Next, a dose-escalation study was performed with the HNSCC cell lines Cal33 and UM-SCC-47 to determine the concentration of nanoparticles at which a toxic effect occurs. A colony formation assay was performed to determine the survival fraction (Figure 4). The surviving fraction of cancer cells is reduced with increasing concentration. High concentrations of Au-SPIONs were more toxic (Figure 4A) compared to Citrate-SPIONs (Figure 4B). Since there should not be a maximum toxicity of the SPIONs for the study of the combined effect of the SPIONs and the ionizing radiation, the concentrations of 2.5 µg/mL Au-SPIONs and 40 µg/mL Citrate-SPIONs were selected.

### 3.3. Methods of Apoptosis, Necrosis, Colony Formation and DNA Double Strand Break Repair

Cell death by apoptosis and necrosis were studied by staining the cells with Annexin V APC and 7 AAD. (Figure 5A). Cells were treated with 2.5 µg/mL Au-SPIONs and 40 µg/mL Citrate-SPIONs. Additionally, colony-forming assays were used (Figure 5B). DNA double-strand breaks (DNA dsbs) were determined by anti-*γ*H2Ax and cell cycle phase by Ki-67 staining (Figure 5C).

### 3.4. Apoptosis and Necrosis Induction by SPIONs and Combined Treatment

Almost all malignant cell lines in both treatments were clearly affected by IR and SPIONs. Apoptosis and necrosis increased in the cell lines UM-SCC-47 (p_Au-SPIONs_ = 0.0476, p_Citrate-SPIONs_ = 0.0238), UD-SCC-2, HSC-4 (p_Citrate-SPIONs_ = 0.0238) and Cal33 for Citrate-SPIONs. To verify an additional effect from the combination of SPIONs and IR, the cytotoxic effect of the nanoparticles was normalized (Figure 6, dashed lines). In the cell lines UM-SCC-47 and HSC-4, a tendency for an increase of cell death occurred when gold nanoparticles (2.5 µg/mL) were combined with ionizing radiation at 2 Gy. Apoptotic and necrotic death increased by a factor of 1.6 in HSC-4 and 1.4 in UM-SCC-47. In cell line Cal33 and UD-SCC-2, apoptosis and necrosis are not enhanced by the combination of Au-SPIONs and ionizing radiation (Figure 6A). Healthy keratinocytes were not additionally affected by the combined treatment, whereas toxicity increased by a factor of 2.8 in the fibroblast cell line. Treatment of all malignant cell lines with Citrate-SPIONs combined with IR induces an additional effect of apoptosis and necrosis. Cell death increased most by a factor of 6.2 for HPV- cell line Cal33. For malignant HSC-4 and UD-SCC-2 cells the factor was found to be 1.7 and 1.4, respectively. UM-SCC-47 cell death increase slightly by a factor of 1.3 For healthy keratinocytes, the combination of Citrate-SPIONs and 2 Gy ionizing radiation leads to an increase by a factor of 1.3. An effect on healthy fibroblast keratinocytes was not detectable (Figure 6B). For both types of SPIONs, it is apparent that the mechanism of their action is independent of HPV status and is rather dependent on other factors.

### 3.5. Survival Fractions Decrease in Colony Formation Assay

Since there are other causes leading to a decrease in viable cells besides the two types of cell death, apoptosis or necrosis, a colony formation assay was performed in addition to flow cytometry. The assay is the gold standard in radiation biology because it is not limited to specific cell deaths, but also includes senescent cells and other types of death. In addition, the cytostatic effect of the treatment can be studied by the change in colony size.

In this study, the colony formation test was modified by using a specially adapted light microscope to count colonies and determine colony size (Figure 7). For most cell lines, the number of colonies and colony size respond similarly. The HNSCC tumor cell lines were distinctly differentially affected. The effect on the studied cell lines can be described in two ways. The cytotoxic effect of the nanoparticles and the 2 Gy ionizing radiation (Figure 7 red line) and the additional effect resulting from the combination of both treatments (Figure 7 red dashed line). Both investigated nanoparticles in combination with irradiation lead to a reduction of survival fraction (SF) and colony size (CS) in the HNSCC cell lines UM-SCC-47 (p_Au-SPIONs (CS)_ = 0.0411, p_Citrate-SPIONs (SF)_ = 0.0044, p_Citrate-SPIONs (CS)_ = 0.0002), UD-SCC-2, HSC-4 (p_Citrate-SPIONs (CS)_ = 0.0003, p_Citrate-SPIONs (SF)_ = 0.0002), Cal33 (p_Au-SPIONs (SF)_ = 0.0002, p_Au-SPIONs (CS)_ = 0.0001, p_Citrate-SPIONs (SF)_ = 0.003, p_Citrate-SPIONs (CS)_ = 0.002). However, an additional combined effect could not be shown in all cell lines. Treatment with Au-SPIONs had the largest additional effect on HPV+ UD-SCC-2 and HPV- Cal33, while the other two tumor cell lines were only slightly affected (Figure 7A). Treatment with Citrate-SPIONs leads to a decrease in cell survival of cell lines UM-SCC-47 (p_Citrate-SPIONs_ = 0.0009) and HSC-4 (p_Citrate-SPIONs_ = 0.0023), while the Cal33 and UD-SCC-2 cell lines were only slightly affected (Figure 7B). Normal skin fibroblasts were marginally affected by IR and there was no additional effect compared with the combination of Au- or Citrate-SPIONs. Keratinocytes were synergistically affected in cell survival and colony size by IR (p_Au-SPIONs (SF)_ = 0.0411, p_Citrate-SPIONs (SF)_ = 0.0097, p_Citrate-SPIONs (CS)_ = 0.0002) and there was an additive effect when IR was combined with SPIONs (p_Au-SPIONs (SF)_ = 0.0152, p_Citrate-SPIONs (SF)_ = 0.0245, p_Citrate-SPIONs (CS)_ = 0.002).

### 3.6. Tumor Cells Effected by IR in Cell Division via Anti-γH2Ax

Based on the question of in which cell mechanism the nanoparticles or combination of nanoparticles with IR interfere, we have performed immunostaining experiments mainly to investigate if increased DNA double-strand breaks are caused by the treatment. Therefore, the HPV-Cal33 cell line was chosen exemplarily. Cal33 has been stained with primary antibodies anti-*γ*H2Ax for DNA double-strand breaks and Ki-67 for cell proliferation. To differentiate between cells that are actively in the cell cycle (growth fraction) and those that are in G0, we used Ki-67 intensity and Ki-67- cells as G0 cells and Ki-67+ cells as growth fraction (Figure 8). Remaining DNA double-strand breaks after a repair of 24 h were identified by anti-*γ*H2Ax. Spontaneous DNA double-stand breaks tended to increase in the G0 phase and clearly increased in the growth fraction of the cells (*p* = 0.02). 2 Gy ionizing radiation increased DNA double-strand breaks distinct, while there was no additional effect by the combination of SPIONs and ionizing radiation.

## 4. Discussion

In this study, we studied the toxicity of Au-SPIONs and Citrate-SPIONs on HNSCC and two healthy tissue cell lines and, in particular, the combined effect of SPIONs and ionizing radiation. Firstly, there must be an interaction between the SPIONs and the cells in order to have an effect [38]. The interactions between nanoparticles and cells are determined by various properties of magnetic nanoparticles, such as size, shape and the functional groups and charges on the particles [39,40]. Often, it is hard to determine if the particles are outside or inside the cell when using light microscopy. It is also possible that different SPIONs, such as Citrate-SPIONs and Au-SPIONs, behave differently when taken up into cells due to their size or other properties. Here, however, both SPION types seem to have a clear affinity for attaching to cells.

The four HNSCC and two healthy tissue lines used responded differently to treatment with SPIONs. What they have in common is that Citrate- and Au-SPIONs have a cytotoxic effect on cells, leading to an increase in cell death and a decrease in colony formation ability. At high concentrations, Au-SPIONs were more cytotoxic than Citrate-SPIONs (Figure 4), which is why concentrations of 2.5 µg/mL Au-SPIONs and 40 µg/mL Citrate-SPIONs have been used to perform the experiments. In a study by Connor et al. it was described that the cytotoxicity of Au-nanoparticles depends on their size and surface area. Three different surface modifiers were used with citrate, biotin and precursor solution, all giving different results in terms of cell survival after three days of exposure. The largest nanoparticles containing citrate and biotin surface modifiers were found to be non-toxic up to a concentration of 250 µM. In contrast, the same concentration of gold salt solution (AuCl_4_) was over 90% toxic. In contrary it was reported that gold (Au)-coated SPIONs were nearly not toxic in an osteoblastic mouse cell line [41]. Overall, it is stated that depending on synthesis and functionalization of SPIONs, the cytotoxicity and metabolism varies, and the loaded cells also show different cellular behaviors.

The main objective was to study cell survival and colony formation between cancer cells and healthy cell tissue. Contrary to expectations, there was an increase in cell death of tumor cells as well as healthy cells. We also expected an additional effect from the combination of SPIONs and irradiation. The cell survival and colony formation assays indicate that SPIONs act very differently on different cell lines. For UM-SCC-47, the combination of SPIONs and IR clearly causes an additional effect. The other HPV+ cell line UD-SCC-2 is less affected by the combination (Figure 6 and Figure 7). Neither HPV status nor tumor cells compared to normal tissue cells are specifically affected by SPIONs or their combination with IR. One possibility for the different effect of the SPIONs could be a different interaction of the SPIONs with the cells or their uptake. Different uptake of SPIONs into cells could ultimately lead to different concentrations of SPIONs in cells and thus different toxicities. To better understand what exactly causes cell death when cells are treated with SPIONs, we captured images of normal skin fibroblasts and HPV-Cal33 under SPION treatment. However, we could not perceive any obvious differences. SPIONs either attach themselves to the cells or are actively incorporated (Figure 2 and Figure 3). In Stein et al. [35] it was described that the uptake of SPIONs in a cell depends on their concentration and size. Connor et al. indicated that nanoparticles are taken up into cell cytoplasm. In their study they show that the concentration of nanoparticles in cell media dropped rapidly within 1 h to a plateau [42]. If there is to be an interaction of SPIONs and IR, DNA should be the target of the combined effect. Unrepaired remaining DNA double-strand breaks lead to cell death. With ionizing radiation, the main mechanism is the induction of DNA double-strand breaks. The mechanism of action of SPIONs is mainly the generation of reactive oxygen species [43]. The release of Fe^2+^ ions and hydrogen peroxide can generate the highly reactive hydroxyl radicals via Fenton type reactions [44,45]. Reactive oxygen species are capable of efficiently damaging molecular structures and can induce DNA damage, including DNA double-strand breaks. Therefore, we performed an immunostaining experiment in which DNA double-strand breaks can be visualized by anti-*γ*H2Ax-immunofluorescence microscopy. Using Ki-67, we differentiated the G0 fraction from the growth fraction undergoing cell division. This is because replication forks that are stalled in the synthesis phase are also labeled with anti-*γ*H2Ax. In the G0 phase, slightly more DNA double-strand breaks were initially present, but significantly more were present in the growth fraction (Figure 8). The initial increased occurrence of DNA double-strand breaks, especially in the growth fraction, indicates that crosslinks stalling replication forks are more likely to occur than direct DNA double-strand breaks.

## 5. Conclusions

Effects induced by the combination of SPIONs and IR were seen to varying degrees in all malignant HNSCC as well as in the healthy cell lines. Overall, the effects triggered by the combination of SPIONs and IR were small. However, in fractionated radiation therapy, even small effects at around 30 fractions can lead to a significant improvement in the end, especially since only very small effects occurred in the cell lines from healthy tissue. SPIONs definitely have a potential to lead to an improvement in radiotherapy.

## Figures and Tables

**Figure 1 bioengineering-09-00806-f001:**
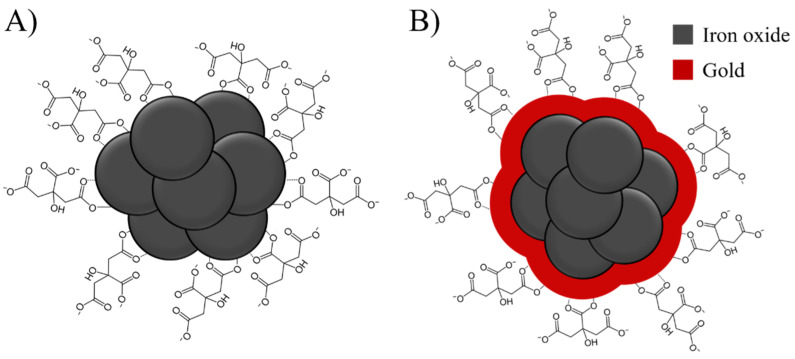
(**A**) Illustration of citrate-stabilized superparamagnetic iron oxide nanoparticles (Citrate-SPIONs); (**B**) gold-coated SPIONs (Au-SPIONs) which are also stabilized using citrate ions.

**Figure 2 bioengineering-09-00806-f002:**
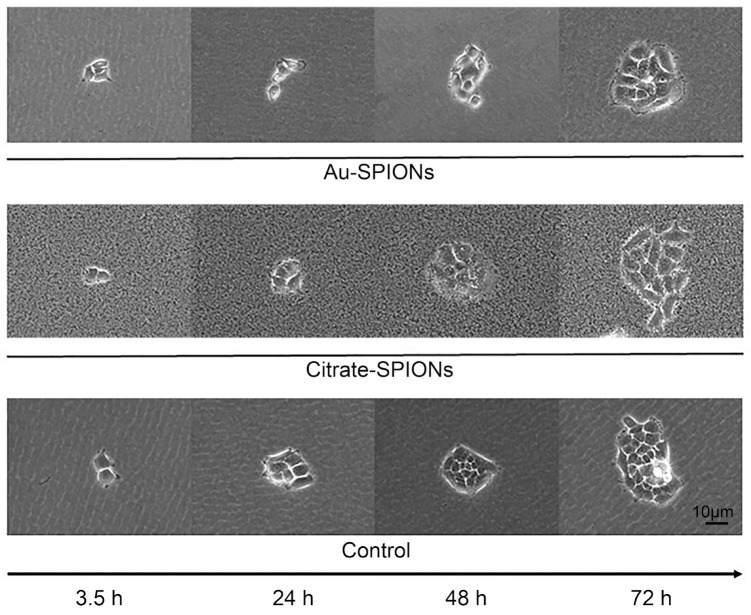
Cal33 HNSCC cells treated with SPIONs and imaged with a light microscope after 3.5 h, 24 h, 48 h and 72 h.

**Figure 3 bioengineering-09-00806-f003:**
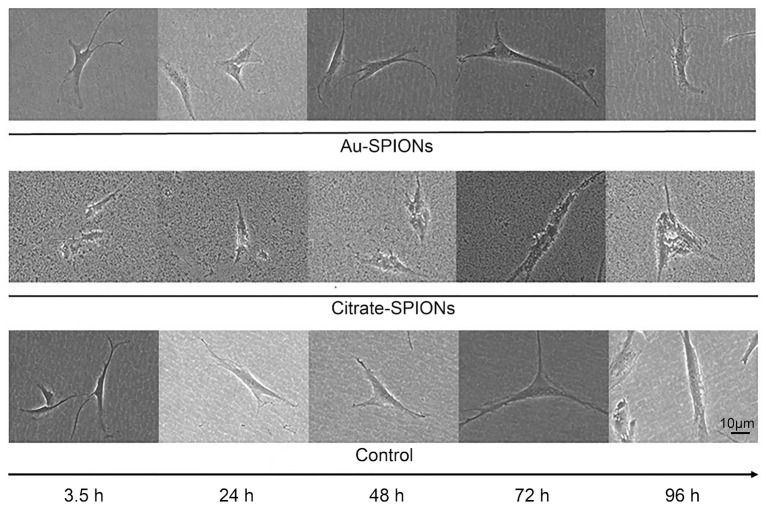
SBLF9 fibroblasts treated with SPIONs and imaged with a light microscope after 3.5 h, 24 h, 48 h, 72 h and 96 h.

**Figure 4 bioengineering-09-00806-f004:**
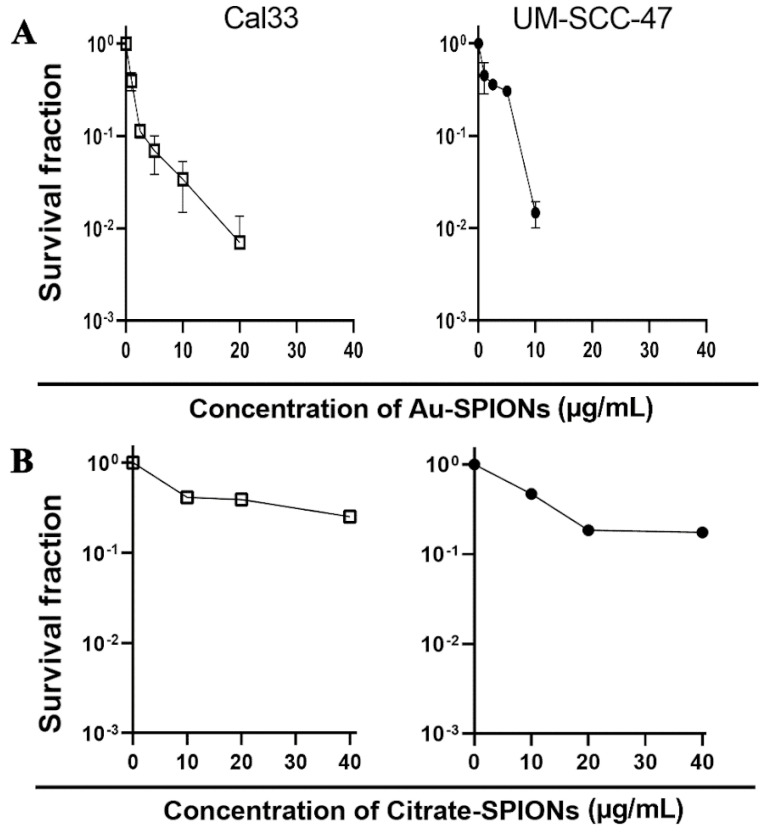
A colony formation assay was performed to determine the toxicity of the SPIONs. HNSCC cell lines Cal33 and UM-SCC-47 were used. Dose escalation for different concentrations of (**A**) Au-SPIONs and (**B**) Citrate-SPIONs.

**Figure 5 bioengineering-09-00806-f005:**
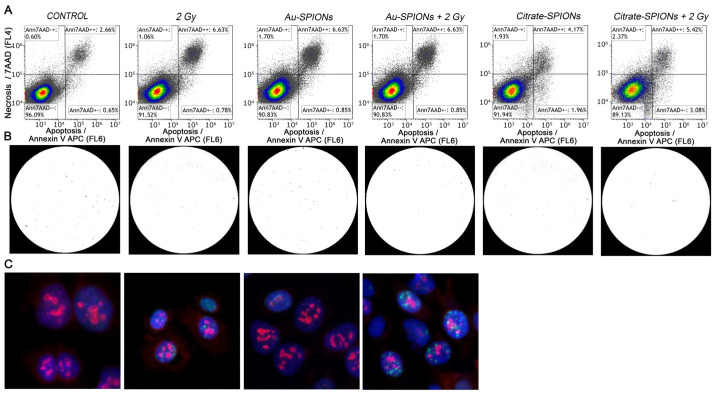
Used methods to detect the combined effect of SPIONs and ionizing radiation. Gating strategy for cell death, colony formation assay and immunofluorescence microscopy. (**A**) Exemplary gating strategy of Annexin-V-APC/7AAD with UM-SCC-47 cells. (**B**) Colony formation assay under microscopy. Images were taken and analyzed with Biomas software. (**C**) Immunofluorescence microscopy of Cal33 tumor cells. Cells were stained with anti-*γ*H2Ax for DNA double-strand breaks (green) and anti-Ki-67 for growth fraction (red). Cell nuclei were stained with DAPI (blue).

**Figure 6 bioengineering-09-00806-f006:**
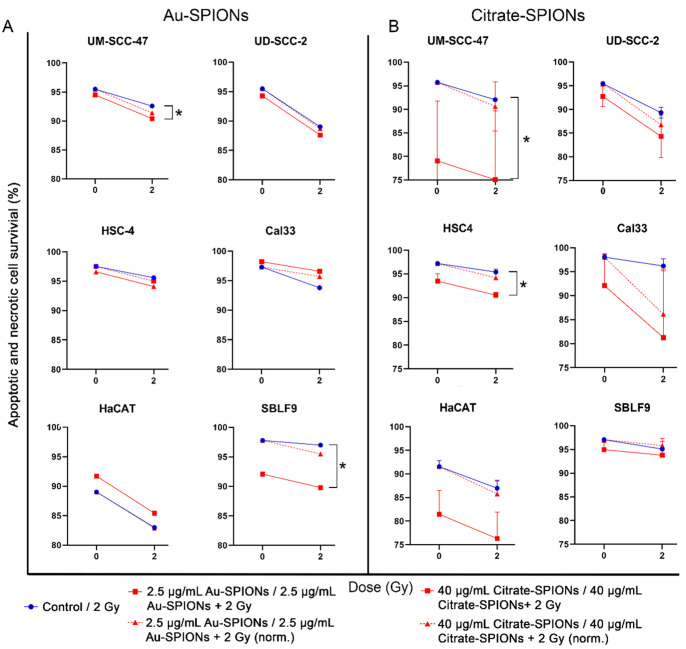
Induction of apoptosis and necrosis were stained with Annexin V APC and 7AAD to determine the differences in cell death of Au- and Citrate-SPIONs in different cell lines. All cell lines were treated with (**A**) Au-SPIONs or (**B**) Citrate-SPIONs. Survival decreased by apoptosis or necrosis under irradiation with 2 Gy (blue line) and a combination of irradiation and Au-SPIONs/Citrate-SPIONs (red line). The red dashed line is normalized to subtract out the cytotoxic effect of the nanoparticles to better indicate any additive effect. The graph depicts HPV+ cell lines in the top row, HPV- cells in the middle row, and healthy cells in the bottom row. Each experiment was repeated at least three times. Error bars indicate the standard deviation. * *p*-value ≤ 0.05.

**Figure 7 bioengineering-09-00806-f007:**
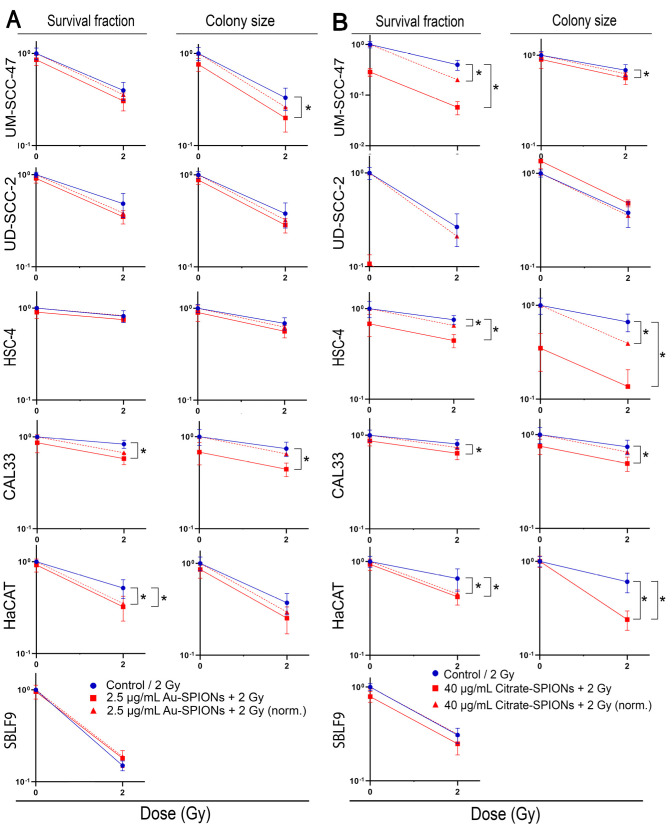
Survival fraction was determined by the colony formation assay. The cells were grouped in HPV+ (UM-SCC-47 and UD-SCC-2), HPV- (HSC-4 and Cal33) and healthy cells. Irradiation was indicated in blue and Au-SPIONs/Citrate-SPIONs in combination was indicated in red. The red dashed line is normalized to subtract out the cytotoxic effect of the nanoparticles to better indicate any additive effect. (**A**) shows graphs for Au-SPIONs and (**B**) Citrate-SPIONs. The graph depicts HPV+ cell lines (UM-SCC-47 and UD-SCC-2) in the two top rows, HPV- cells in the two middle rows (HSC-4 and Cal33), and healthy cells in the two bottom rows. Each experiment was repeated at least three times. Error bars indicate the standard deviation. * *p*-value ≤ 0.05.

**Figure 8 bioengineering-09-00806-f008:**
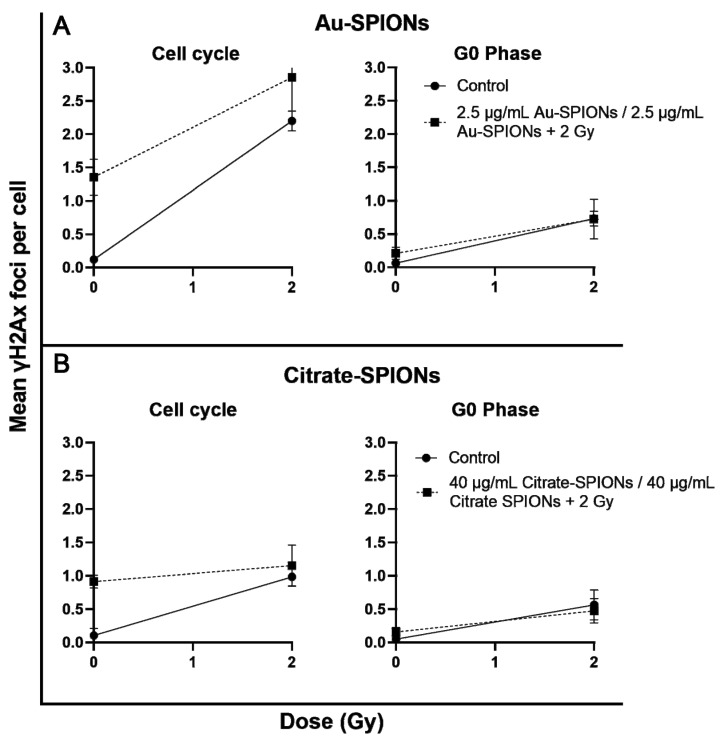
Anti-*γ*H2Ax foci as DNA double-strand breaks in the presence of Au- and Citrate-SPIONs. Cal33 cells were differentiated into G0 phase and growth fraction by Ki-67. Cells were (**A**) treated with Au-SPIONs or (**B**) Citrate-SPIONs. DNA-double strand breaks were identified after a repair time of 24 h with or without 2Gy ionizing radiation. Each experiment was repeated at least three times. Error bars indicate the standard deviation.

## Data Availability

The data presented in this study are available on request from the corresponding author.

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
