# Peer review of "Effect of Citrate- and Gold-Stabilized Superparamagnetic Iron Oxide Nanoparticles on Head and Neck Tumor Cell Lines during Combination Therapy with Ionizing Radiation"

_bioengineering, 2022, doi:10.3390/bioengineering9120806_

Round 1

Reviewer 1 Report

In this work, two types of nanoparticles, citrate-coated iron oxide nanoparticles (citrate-SPIONs) as well as gold/citrate-coated iron oxide nanoparticles (Au-SPIONs) were employed for studying their uptake and irradiation effects on fibroblast and kerarinocytes.

The characterization data (e.g., TEM, EDX, DLS, etc.) of both nanoparticles (citrate-SPIONs and Au-SPIONs) must be provided, to prove the morphology, size and purity of the products.

Prussian blue staining can be performed to observe the localization of the nanoparticles within cells.

The reference numbers in the main text were not matched with the reference session. For example, Stein et al [30],  Elbialy et al [31] and Mühlberger et al. [32] were not cited in order.

The authors may have overlooked some previously published articles on the related research directions. For example:

1.     Leung, K. C.-F. et al. Citrate-Coated Magnetic Polyethyleneimine Composites for Plasmid DNA Delivery into Glioblastoma. Polymers 2021, 13, 2228 and the references therein.

2.     Cucci, L.M. et al. Gold nanoparticles functionalized with angiogenin for wound care application. Nanomaterials 2021, 11, 201 and the references therein.

Reviewer 2 Report

The manuscript contains many professional results on biomedical experiments, which is beyond the bounds of my ability. I can not propose too much comments.

Just one, the caption for Figure 5 is  too short or not detailed.

Author Response

Dear editor, dear reviewers,

Thank you for the constructive criticism. We edited our manuscript according to your recommendations point by point. In the following, your comments are printed in italics and the insertions in the manuscript in blue. We hope our manuscript is now appropriate for publication in “Bioengineering”

comment: The manuscript contains many professional results on biomedical experiments, which is beyond the bounds of my ability. I can not propose too much comments.

Just one, the caption for Figure 5 is  too short or not detailed.

Answer: Very many thanks for the review! We have clarified the figure caption and described it in more detail.

Reviewer 3 Report

Effect of citrate- and gold-stabilized superparamagnetic iron oxide nanoparticles on head and neck tumor cell lines during combination therapy with ionizing radiation

The manuscript has been written and presented well, however, the following points should be addressed by authors

ABSTRACT Line 1

Abstract: (1) HNSCC is the sixth most common cancer worldwide.

What is meaning of (1) here. It seems that it is reference number.

Please remove references from abstract.

Also expand the abbreviation.

Or better avoid it in the abstract.

There are other abbreviations also which should be expanded.

Page 2 first paragraph

Chemotherapies in particular often have a number of side effects that can lead to serious complications [10, 11]. But surgeries are also associated with risks [12]. For example, in the form of wound healing problems [13].

Sentences are vague.

Please simplify all sentences and add specific side effect.

Page 3 last paragraph

2.3. Flow cytometry analysis of apoptosis and necrosis.

The second group was treated with 10 µg/mL Citrate-SPIONs

Please elaborate on dose.

Was it a theoretical dose (based on how much you added in reaction mixtures)

Or based on the practical yield value / entrapped amount?

Please include similar elaboration for the dosing of 2.5 µg/mL Au-SPIONs

Page 13 Discussion, paragraph 3

The main objective was to study cell survival and colony formation between cancer cells and healthy cell tissue. Contrary to expectations, there was an increase in cell death of tumor cells as well as healthy cells. We also expected an additional effect from the combination of SPIONs and irradiation. The cell survival and colony formation assays indicate that SPIONs act very differently on different cell lines. For UM-SCC-47, the combination of SPIONs and IR clearly causes an additional effect. The other HPV+ cell line UD-SCC-2 is less affected by the combination.

Please add results or refer section / figure to support the discussion wherever required in the discussion part.

Page 15

Conclusion is missing

Please incorporate the conclusions of the study.
